# Assessing perioperative risks in a mixed elderly surgical population using machine learning: A multi-objective symbolic regression approach to cardiorespiratory fitness derived from cardiopulmonary exercise testing

**Pietro Arina** [1,2]*, **Davide Ferrari** [3], **Maciej R. Kaczorek** [4], **Nicholas Tetlow** [2], **Amy Dewar** [2], **Robert Stephens** [2], **Daniel Martin** [5], **Ramani Moonesinghe** [2], **Mervyn Singer** [1], **John Whittle** [2�], **Evangelos B. Mazomenos** [4�]

1 Bloomsbury Institute of Intensive Care Medicine, University College London, London, United Kingdom, 2 Human Physiology and Performance Laboratory, Centre for Perioperative Medicine, Department of Targeted Intervention, University College London, London, United Kingdom, 3 Department of Population Health Sciences, King's College London, London, United Kingdom, 4 Wellcome/EPSRC Centre of Interventional and Surgical Sciences and Department of Medical Physics and Biomedical Engineering, University College London, London, United Kingdom, 5 Peninsula Medical School, University of Plymouth, Plymouth, Devon, United Kingdom

ो These authors contributed equally to this work and share senior authorship.
* p.arina@ucl.ac.uk

## Abstract

Accurate preoperative risk assessment is of great value to both patients and clinical teams. Several risk scores have been developed but are often not calibrated to the local institution, limited in terms of data input into the underlying models, and/or lack individual precision. Machine Learning (ML) models have the potential to address limitations in existing scoring systems. A database of 1190 elderly patients who underwent major elective surgery was analyzed retrospectively. Preoperative cardiorespiratory fitness data from cardiopulmonary exercise testing (CPET), demographic and clinical data were extracted and integrated into advanced machine learning (ML) algorithms. Multi-Objective-Symbolic-Regression (MOSR), a novel algorithm utilizing Genetic Programming to generate mathematical formulae for learning tasks, was employed to predict patient morbidity at Postoperative Day 3, as defined by the PostOperative Morbidity Survey (POMS). Shapley-Additive-exPlanations (SHAP) was subsequently used to analyze feature contributions. Model performance was benchmarked against existing risk prediction scores, namely the Portsmouth-Physiological-and-Operative-Severity-Score-for-the-Enumeration-of-Mortality-and-Morbidity (PPOSSUM) and the Duke-Activity-Status-Index, as well as linear regression using CPET features. A model was also developed for the same task using data directly extracted from the CPET time-series. The incorporation of cardiorespiratory fitness data enhanced the performance of all models for predicting postoperative morbidity by 20% compared to sole reliance

**Data availability statement:** The data underlying this paper is sensitive and involves patient information, and public deposition would compromise patient privacy, which is a breach of the protocol approved by the research ethics board. To ensure compliance with ethical and legal standards, we have made the data accessible through a controlled process. Interested researchers can request access by contacting the appropriate personnel via the address provided on our research webpage https://www.uclh.nhs.uk/research.

**Funding:** This work was supported by the Cleveland Clinic London Hospital, London, UK, and the Mittal Fund at Cleveland Clinic Philanthropy (UK) (to PA); by King's College London and DRIVE-Health, a KCL-funded Centre for Doctoral Training in Data-Driven Health (to DF); by the University College London Hospitals National Institute of Health Research Biomedical Research Centre Critical and Perioperative Care theme and in part by an International Anaesthesia Research Society Mentored Research Grant (to JW); and by the Wellcome/EPSRC Centre for Interventional and Surgical Sciences at University College London (to EM). The funders had no role in study design, data collection and analysis, decision to publish, or preparation of the manuscript.

**Competing interests:** The authors have declared that no competing interests exist.

on clinical data. Cardiorespiratory fitness features demonstrated greater importance than clinical features in the SHAP analysis. Models utilizing data taken directly from the CPET time-series demonstrated a 12% improvement over the cardiorespiratory fitness models. MOSR model surpassed all other models in every experiment, demonstrating excellent robustness and generalization capabilities. Integrating cardiorespiratory fitness data with ML models enables improved preoperative prediction of postoperative morbidity in elective surgical patients. The MOSR model stands out for its capacity to pinpoint essential features and build models that are both simple and accurate, showing excellent generalizability.

## Author summary

Accurately predicting postoperative complications in elderly patients undergoing surgery remains a significant challenge. Traditional risk scores often fail to account for key physiological indicators such as cardiorespiratory fitness, which can provide valuable insights into a patient's overall health and resilience. In this study, we explored whether integrating cardiorespiratory fitness data into machine learning models could improve the prediction of postoperative morbidity. We introduced a novel approach using Multi-Objective Symbolic Regression (MOSR), an interpretable machine learning technique that balances accuracy with simplicity. Our results show that models incorporating fitness data—particularly those using MOSR—significantly outperformed both conventional risk scores and other machine learning models. These findings highlight the value of combining domain-specific physiological data with advanced analytics to enhance clinical decision-making. By improving risk stratification, this approach has the potential to support more personalized peri-operative care and better outcomes for elderly surgical patients.

## Introduction

As the number of operative procedures carried out on a population that is becoming more comorbid and frailer grows [1,2], so does the incidence of postoperative morbidity and mortality [3–8]. Accurate preoperative risk prediction supports patients in making informed decisions and guides clinical decision-making. Existing risk scores include the Portsmouth-Physiological and Operative Severity Score for the enumeration of Mortality and Morbidity (PPOSSUM) [9]. However, such scores are limited by their reliance on linear regression models that often omit or barely incorporate physiological data such as measures of cardiorespiratory fitness (CRF) that can predict postoperative outcomes [10].

Cardiopulmonary Exercise Testing (CPET) is the gold standard for preoperative assessment of CRF. Several headline parameters are commonly applied to

preoperative outcome prediction such as ventilatory anaerobic threshold (AT) and peak oxygen consumption ($\dot{V}O_2$) (Fig 1A) [11–16]. Inclusion of CPET variables into risk prediction models has also been limited in scope. In addition, the complexity of time-series physiological data has prevented large scale analysis. Mechanisms through which CRF affects postoperative outcomes are under investigation [13]. There is a distinct need to better characterize the utility of CPET, both alone and alongside other tools used for multifactorial risk stratification.

Machine Learning (ML) models utilizing demographic and clinical data have been applied extensively in perioperative medicine [17,18]. However, no ML models have yet incorporated CRF data or focused exclusively on preoperative information [18]. We aimed to identify the optimal ML algorithm that utilizes CRF data for predicting postoperative outcomes. One option we chose to explore was the Multi-objective Symbolic Regression (MOSR) approach [19,20], an algorithm

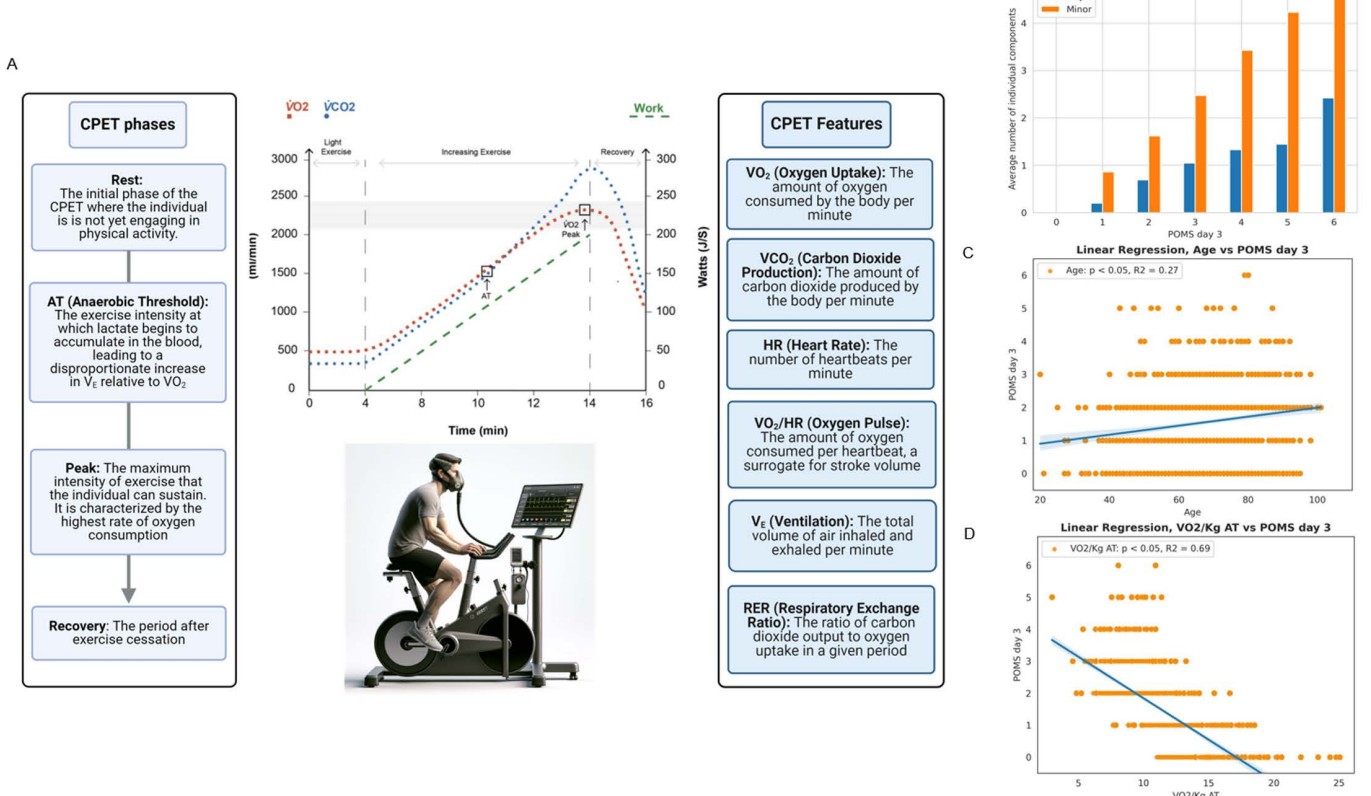

**Fig 1. A - Cardiopulmonary exercise testing (CPET).** On the left, the different phases of CPET are detailed, including Rest, Ventilatory Anaerobic Threshold (AT), Peak, and Recovery, describing the exercise intensity levels and physiological responses. The central graph displays the progression of exercise intensity, marked by the green ramp, alongside traces of oxygen uptake (VO2) and carbon dioxide production (VCO$_2$). The right section elucidates primary features derived from CPET such as VO2, VCO$_2$, heart rate (HR), oxygen pulse (VO$_2$/HR), ventilation (VE), and the respiratory exchange ratio (RER). The image portrays a patient undergoing CPET, equipped with the appropriate testing apparatus. The graph is presented with the consent of Quick O, Reed-Poysden C, from their work "Cardiopulmonary Exercise Test: Interpretation and Application in Perioperative Medicine" 2022, https://doi.org/10.28923/atotw.473. **B -** Bar chart illustrating the average number of minor and major components contributing to each Postoperative Morbidity Survey (POMS) score on day 3. The bars represent the mean count of individual minor (blue) and major (orange) morbidity factors for patients with POMS scores ranging from 0 to 6. **C -** Scatter plot with linear regression analysis illustrating the relationship between patient age and Postoperative Morbidity Survey (POMS) scores on day 3. Each datapoint represents an individual patient's age against their corresponding POMS score. **D -** Scatter plot with linear regression analysing the relationship between VO2/Kg at the Anaerobic Threshold (AT) and Postoperative Morbidity Survey (POMS) scores on day 3. Each datapoint signifies the VO2/Kg AT value for an individual patient and their respective POMS score.

based on genetic programming that generates a set of readable mathematical equations that can be used as predictive models [20].

We hypothesized that applying ML to perioperative data, particularly incorporating preoperative CPET-derived CRF, would accurately predict postoperative morbidity in a population of elderly patients with multiple comorbidities undergoing major surgery. Our goals were to assess the impact of CPET and preoperative electronic healthcare record (EHR) data in a broad perioperative elective surgical fragile patient dataset on predicting postoperative outcomes, and to compare the effectiveness of MOSR against current machine learning algorithms and clinical risk scores.

## Methods

University College London Hospitals NHS Foundation Trust (UCLH) maintains a prospective research database of patients undergoing CPET before major surgery. All participants provided written consent for their CPET outcomes to be included in the database for future research, in compliance with the Declaration of Helsinki. Ethical approval was initially granted in 2012, and reaffirmed in 2019, with no specified time constraints (NRES Committee London – Southeast reference: 12/LO/0192, London - Westminster Research Ethics Committee reference: 19/LO/1371). The database was created to collect data for assessing and studying short- and long-term postoperative morbidity and mortality, encompassing a broad range of patients who underwent CPET prior to major surgery. The initial analysis was performed by two clinical exercise physiologists and was later confirmed by a consultant anaesthetist. The database was queried for patients enrolled between 2012 and 2022. Adherence to Caldicott principles ensured data confidentiality and proper collection methods.

### Population and selection criteria

Eligible participants over 18 years routinely referred for CPET as part of their preoperative assessment and listed for elective major non-cardiac surgery at University College London Hospitals NHS Foundation Trust were included. Patients under 18 years or unable to provide consent were excluded. CPET was performed in accordance with International Peri-Operative Exercise Testing and Training Society (iPOETTS) guidance. Where iPOETTS medical criteria precluded testing, patients were excluded [21]. Of 2,145 patients assessed, 1,190 were included in the tabular database, and 585 with full CPET recordings for time-series analysis (CONSORT flow diagram; S1 Fig). Table 1 contains relevant demographic data.

### Database

The main dataset was divided into four components. A full feature list in provided in the Appendix:

(i)  Clinical dataset (39 features) with demographics and medical history – age, biometrics, medical and drug history, laboratory values, and surgery details (including severity and technique).

(ii)  CRF dataset (46 features) - derived cardiorespiratory fitness values were extracted during different phases of the CPET (Fig 1A). Measurements were made of minute oxygen uptake ($\dot{V}O_2$), minute carbon dioxide production ($\dot{V}CO_2$), and end-tidal gas tensions ($P_{ET}O_2$ and $P_{ET}CO_2$) during a ramped exercise protocol performed on a cycle ergometer. Ventilatory equivalents for oxygen (the slope between minute ventilation and oxygen uptake: $V_E/\dot{V}O_2$) and carbon dioxide (the slope between minute ventilation and carbon dioxide production: $V_E/\dot{V}CO_2$), and oxygen pulse (a correlate of stroke volume, the slope between oxygen uptake and heart rate (HR): $\dot{V}O_2/HR$) were derived. $\dot{V}O_2$ peak was defined as the highest average $\dot{V}O_2$ over the last 30 seconds of ramped exercise. The ventilatory anaerobic threshold (AT) was determined using the V-slope method, ventilatory equivalents, and end-tidal gas tensions [21]. Two clinical exercise physiologists independently interpreted the tests and then subsequently verified by a consultant anesthetist.

**Table 1.  Population demographics, comorbidities, laboratory variables, medications and surgery types, cardiopulmonary exercise test (CPET) values and perioperative outcomes.**

| Demographic | Median (IQR) |
| --- | --- |
| Age (years) | 71 (61-79) |
| Sex (M/F) | 825/365 – 69%/31% |
| BMI (kg/m$^2$) | 26.5 (23.3-30) |
| ASA | 2 (2-3) |
| DASI | 46.2 (32.2-58.2) |
| P-POSSUM (operative score) | 14 (12-16) |
| P-POSSUM (physiological score) | 18 (16-21) |
| **Comorbidities** | **N (%)** |
| Hypertension | 416 (35%) |
| Diabetes | 131 (11%) |
| Angina | 48 (4%) |
| Coronary stent | 60 (5%) |
| Previous coronary artery bypass graft (CABG) | 36 (3%) |
| Chronic cardiac failure | 119 (10%) |
| Peripheral vascular disease | 24 (2%) |
| Previous cerebrovascular accident (CVA) or transient ischaemic attack (TIA) | 47 (4%) |
| Chronic obstructive pulmonary disease | 72 (6%) |
| Asthma | 95 (8%) |
| Previous pulmonary embolism | 17 (2%) |
| Pulmonary fibrosis | 10 (1%) |
| Smoking (former/current) | 328 (27%) |
| **Laboratory values** | **Median (IQR)** |
| Na (mmol/L) | 141 (139-142) |
| K (mmol/L) | 4.3 (4.1-4.6) |
| Creatinine (µmol/L) | 79 (66-95) |
| Urea (mmol/L) | 6.1 (4.6 - 7.5) |
| White blood cells (x 10$^9$/L) | 7.7 (6.2-8.8) |
| Haemoglobin (g/L) | 131 (119-142) |
| eGFR (ml/min) | 84 (68-114) |
| **Medications** | **N (%)** |
| Beta blocker | 238 (20%) |
| Nitrate | 36 (3%) |
| ACE Inhibitor | 214 (18%) |
| Statin | 357 (30%) |
| **Operation Type (%)** | **N (%)** |
| 1 – Colorectal | 244 (20%) |
| 2 – Upper Gastrointestinal | 196 (17%) |
| 3 – Genitourinary | 404 (34%) |
| 4 – Head and Neck | 253 (21%) |
| 5 – Thoracic | 43 (4%) |
| 6 – Others | 50 (4%) |
| **CPET Values** | **Median (IQR)** |
| Systolic arterial pressure (mmHg) | 131 (121-145) |
| Diastolic arterial pressure (mmHg) | 79 (70-87) |

*(Continued)*

**Table 1.** (Continued)

| Demographic | Median (IQR) |
|---|---|
| Mean arterial pressure (mmHg) | 97 (88-105) |
| $SpO_2$ (%) | 96 (96-97) |
| MET | 4.5 (3.7-5.5) |
| **CPET – Baseline (rest) values** | |
| HR (bpm) | 82 (73-93) |
| $\dot{V}O_2$/kg (ml/kg/min) | 3.8 (3.3-4.4) |
| RER | 0.82 (0.78-0.87) |
| $V_E/\dot{V}CO_2$ (ml/min) | 31 (28-33) |
| **CPET – values at AT** | |
| HR (bpm) | 106 (95-118) |
| $\dot{V}O_2$/kg (ml/kg/min) | 10.6 (9-12.4) |
| RER | 0.85 (0.8-0.9) |
| $V_E/\dot{V}CO_2$ (ml/min) | 33 (29-36) |
| **CPET – Peak values** | |
| HR (bpm) | 136 (120-152) |
| $\dot{V}O_2$/kg (ml/kg/min) | 16.9 (13.9-20.5) |
| RER | 1.1 (1.0-1.2) |
| $V_E/\dot{V}CO_2$ (ml/min) | 34 (30-37) |
| **Outcome** | N (%) |
| 1 year mortality | 66 (5.5%) |
| 30 days mortality | 23 (1.9%) |
| Readmission at 30 days | 88 (7.3%) |
| Adverse event | 178 (14.9%) |
| Post operative destination (Ward/High Dependency Unit (HDU)/Intensive Care Unit (ICU)) | 345 (29%), 762 (64%), 83 (7%) |
| Length Of Stay | 10 (7-16) |
| POMS day 3 | 3 (2-4) |
| POMS day 5 | 2 (1-3) |
| POMS day 7 | 2 (1-3) |

BMI: Body Mass Index, ASA: American Society of Anaesthesiology grade, DASI: Duke Activity Status Index, PPOSSUM: Portsmouth – Physiological and Operative Severity Score for the enumeration of Mortality and morbidity, $SpO_2$:Oxygen saturation, MET: metabolic equivalents of task, VO2/Kg: rate of oxygen consumption per kilogram, $V_E/\dot{V}CO_2$: ventilatory equivalents for $CO_2$, HR: heart rate, RER: Respiratory Exchange Ratio, AT: Ventilatory Aerobic threshold in ml $O_2$/kg/min, POMS: Postoperative Morbidity Survey, Na: Sodium, K: Potassium, eGFR: estimated Glomerular Filtration Rate. Values reported as median with interquartile range (IQR) or absolute number and percentage over total (%).

$VO_2$ Peak and Anaerobic Threshold (AT) values were adjusted for body weight ($ml.min^{-1}.kg^{-1}$). Electrocardiographic and expired gas data were collected at intervals of one sample per second and median filtered for each patient.

(iii) CRF-TS (time-series) (18 features). The dataset of a subset of 585 patients was extracted from the CRF database. This dataset contained CPET time-series data for 15 features routinely recorded during the CPET exam (VO2, $VCO_2$, etc.), with a resolution of one value per second.

(iv) Outcomes and clinical scores - American Society of Anesthesiologists (ASA) score, Duke Activity Status Index (DASI) score, post-operative care destination, length of hospital stay, mortality and morbidity (as described below) were recorded onto the database.

 

Data were imported from Microsoft Excel read-only files on secured university storage. Routine laboratory results were extracted from the hospital electronic healthcare record system (EPIC, Verona, Wisconsin, USA). Dataset quality was verified by three physicians.

## Major and minor POMS morbidity classification

To evaluate postoperative morbidity, we employed the Post-Operative Morbidity Survey (POMS) on Postoperative Days 3, 5, and 7. POMS is straightforward approach for detecting and quantifying postoperative complications. Designed to be applicable across all types of surgery, its focus is on identifying complications that could hinder a patient's discharge from the hospital. This method has been previously validated within complex surgical populations, underscoring its reliability [22]. POMS was prospectively collected in person by a trained researcher in a controlled hospital environment. The POMS score was calculated as a binary, non-weighted result for any positive score across the nine POMS domains, as outlined in S1 Table. The POMS domains were also categorized into major and minor complications, as reported in literature [23]. Routine care elements such as the presence of nasogastric tubes after upper gastrointestinal surgery or urinary catheters post-cystectomy (also classified as minor POMS) were not classified as morbidities and these domains were excluded from our analysis [23].

Based on initial analysis, the patient population could be classified into two distinct groups (Fig 1B) and this was instrumental for subsequent model development. This classification emerged from the priori observation of the fact major and minor factors influencing the POMS score on day 3 post-surgery, leading to the identification of 557 patients with a day 3 POMS score between 0 and 1 (POMS 0-1), indicative of no or minor morbidities, and 633 patients with a score equal to or above 2 (POMS ≥2), signifying moderate or severe postoperative complications. The decision to prioritize model development on Postoperative Day 3 was influenced by the findings presented in S2 Fig. This figure, produced using linear regression analysis, demonstrated a high correlation between the count of POMS-positive domains on Day 3 and the scores on Days 5 and 7. This approach was further justified by previous reports that the trajectory of postoperative POMS is indicative of 5-year mortality rates [5].

## Statistical analysis

Data analysis and statistical computations were conducted using Python (version 3.10.12) [24] and Pandas (version 1.4.2). Binary encoding was applied to categorical variables, assigning 1 for the occurrence of an event and 0 for its absence. The categorical attributes of sex, surgery type, and specialty were subjected to one-hot encoding due to their non-ordinal nature. Continuous variables underwent normalization to fit within a 0-1 scale. Assessment of data normality was conducted, with results presented either as median with interquartile range or as mean with standard deviation, depending on the distribution. Frequency distributions were evaluated for categorical data. To discern differences between groups, statistical tests such as Student's t-test, Mann-Whitney U test, Chi-Squared test, and Fisher's Exact Test were utilized to analyse the demographic. To assess the relationship between features and outcome, a linear Pearson correlation analysis was performed. A threshold of $p < 0.05$ was established for statistical significance.

## MORS algorithm and classification models

ML classification models were developed based on the POMS classification to predict patients with POMS 0-1 or POMS ≥2 at day 3, utilizing only features available before surgery. Model performances are reported in Fig 2 and Table 2. The initial 80:20 random split was applied to the entire dataset of 1,190 patients, creating two primary subsets: a training set (952 samples) and a test set (238 samples), while maintaining outcome prevalence across groups. This 80/20 division follows standard machine learning practices to ensure a balance between training and evaluation data.

To train the model and optimize its hyperparameters, we conducted a 10-fold cross-validation grid search on the training set. In each iteration, a 90:10 random split was performed, assigning 857 samples for training and 95 for validation. This approach, widely used in machine learning, allows us to systematically evaluate different parameter combinations by assessing model performance on the validation set, leading to the selection of the best hyperparameters.

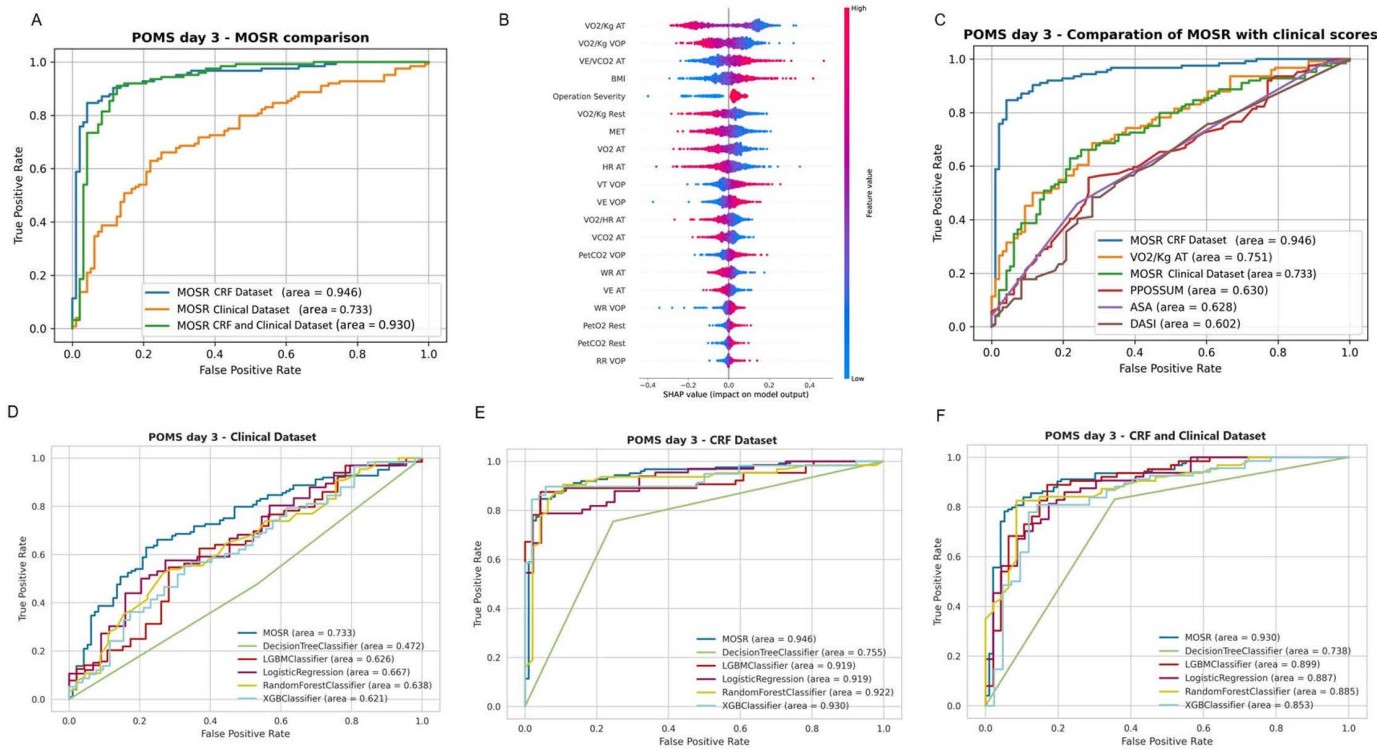

**Fig 2. A - Receiver operating characteristic (ROC) curve comparison for the Multi-objective Symbolic Regression (MOSR) model's performance in predicting Postoperative Morbidity Survey (POMS) scores at day 3.** The ROC curves of three models are presented: MOSR using Cardiorespiratory Fitness (CRF) dataset (blue), MOSR using Clinical dataset (orange), and MOSR utilizing a combination of both CRF and Clinical dataset (green). **B** - SHAP Beeswarm Plot of MOSR models using Cardiorespiratory Fitness (CRF) and Clinical database. Each row represents a feature used in the model. Each dot on a row corresponds to a datapoint, with its position on the x-axis indicating the SHAP value, or the contribution of that feature to the model's prediction for that datapoint. The colour of the dots represents the value of that feature, with blue indicating low values and red indicating high values. $VO_2$/Kg AT: Oxygen Consumption per Kilogram at Anaerobic Threshold, VO2/Kg VOP: Oxygen Consumption per Kilogram at Peak, $V_E$/$VCO_2$ AT: Ventilatory Equivalent for Carbon Dioxide at Anaerobic Threshold, BMI: Body Mass Index, $VO_2$/Kg Rest = Oxygen Consumption per Kilogram at Rest, MET = Metabolic Equivalent of Task, $VO_2$ AT = Oxygen Consumption at Anaerobic Threshold, HR AT = Heart Rate at Anaerobic Threshold, VT VOP = Tidal Volume at Peak, VE VOP = Ventilation at Peak, VO2/HR AT = Oxygen Consumption per Heart Rate at Anaerobic Threshold, $VCO_2$ AT = Carbon Dioxide Production at Anaerobic Threshold, $PetCO_2$ VOP = Partial End-tidal Carbon Dioxide at Peak, WR AT = Work Rate at Anaerobic Threshold, VE AT = Ventilation at Anaerobic Threshold, WR VOP = Work Rate at Peak, $PetCO_2$ Rest = Partial End-tidal Carbon Dioxide at Rest, RR VOP = Respiratory Rate at Peak. **C -** Receiver Operating Characteristic (ROC) curves comparing the Multi-objective Symbolic Regression (MOSR), utilizing either CRT data alone or with Clinical data, against established clinical assessment scores in predicting Postoperative Morbidity Survey (POMS) outcomes on day 3. MOSR CRF dataset model (blue), Cardiopulmonary Exercise Testing (CPET) (red), American Society of Anaesthesiologists Score (ASA) (yellow), Physiological and Operative Severity Score for the enumeration of Mortality and morbidity (PPOSSUM) (purple), and Duke Activity Status Index (DASI) (green). **D–F** - Receiver Operating Characteristic (ROC) curves comparing the Multi-objective Symbolic Regression (MOSR) model, against DecisionTree Classifier, LGBM Classifier, Logistic Regression, Random Forest Classifier, and XGB Classifier. Panel D shows ROC curves for models using only Clinical dataset, Panel E displays ROC curves for models using only Cardiorespiratory Fitness dataset and Panel F presents ROC curves for models using both CRF and Clinical data.

After finalizing the model's parameters and weights, we evaluated its predictive performance and robustness through 10 independent test runs. Each test run utilized a randomly selected 90% subset (214 samples) of the test set (238 samples), and we report the average accuracy obtained across these runs.

Data leakage was not present in our study, as the training and testing sets were strictly partitioned at the outset using an 80:20 random split. The test set (238 samples) remained completely independent and was not utilized during model training, hyperparameter optimization, or cross-validation. This strict separation ensures that model performance was evaluated exclusively on unseen data, maintaining the validity and reliability of the results.

**Table 2. Machine learning model performances.**

| Model – CRF and Clinical Dataset | Accuracy | ±CI | AUC | ±CI | F1 Score | ±CI | Sensitivity | ±CI | Specificity | ±CI | PPV (precision) | ±CI | NPV | ±CI |
|---|---|---|---|---|---|---|---|---|---|---|---|---|---|---|
| MOSR | 0.859 | ±0.025 | 0.930 | ±0.027 | 0.879 | 0.0278 | 0.911 | ±0.035 | 0.850 | ±0.025 | 0.892 | ±0.035 | 0.911 | ±0.035 |
| Decision Tree Classifier | 0.845 | ±0.027 | 0.726 | ±0.026 | 0.811 | 0.0268 | 0.849 | ±0.043 | 0.874 | ±0.023 | 0.840 | ±0.034 | 0.849 | ±0.043 |
| LGBM Classifier | 0.895 | ±0.032 | 0.908 | ±0.029 | 0.805 | 0.0297 | 0.851 | ±0.041 | 0.722 | ±0.036 | 0.800 | ±0.049 | 0.821 | ±0.041 |
| Logistic Regression | 0.859 | ±0.033 | 0.878 | ±0.029 | 0.777 | 0.0293 | 0.865 | ±0.047 | 0.833 | ±0.040 | 0.811 | ±0.064 | 0.815 | ±0.047 |
| Random Forest Classifier | 0.824 | ±0.037 | 0.865 | ±0.030 | 0.700 | 0.0356 | 0.837 | ±0.050 | 0.715 | ±0.032 | 0.812 | ±0.042 | 0.787 | ±0.050 |
| XGB Classifier | 0.898 | ±0.030 | 0.887 | ±0.031 | 0.709 | 0.0277 | 0.825 | ±0.036 | 0.724 | ±0.035 | 0.803 | ±0.047 | 0.825 | ±0.036 |
| | | | | | | | | | | | | | | |
| Model – CRF Dataset | Accuracy | ±CI | AUC | ±CI | F1 Score | ±CI | Sensitivity | ±CI | Specificity | ±CI | PPV (precision) | ±CI | NPV | ±CI |
| MOSR | 0.873 | ±0.014 | 0.946 | ±0.012 | 0.890 | ±0.143 | 0.920 | ±0.052 | 0.864 | ±0.021 | 0.887 | ±0.026 | 0.919 | ±0.035 |
| Decision Tree Classifier | 0.812 | ±0.034 | 0.755 | ±0.032 | 0.832 | ±0.036 | 0.827 | ±0.066 | 0.840 | ±0.024 | 0.793 | ±0.043 | 0.827 | ±0.066 |
| LGBM Classifier | 0.884 | ±0.025 | 0.919 | ±0.024 | 0.796 | ±0.025 | 0.821 | ±0.052 | 0.803 | ±0.025 | 0.774 | ±0.038 | 0.811 | ±0.052 |
| Logistic Regression | 0.862 | ±0.018 | 0.919 | ±0.025 | 0.780 | ±0.017 | 0.893 | ±0.041 | 0.768 | ±0.024 | 0.722 | ±0.043 | 0.833 | ±0.041 |
| Random Forest Classifier | 0.883 | ±0.023 | 0.922 | ±0.022 | 0.794 | ±0.024 | 0.885 | ±0.058 | 0.807 | ±0.024 | 0.779 | ±0.041 | 0.825 | ±0.058 |
| XGB Classifier | 0.880 | ±0.031 | 0.930 | ±0.026 | 0.792 | ±0.029 | 0.881 | ±0.051 | 0.807 | ±0.037 | 0.779 | ±0.056 | 0.811 | ±0.051 |
| Model – Clinical Dataset | Accuracy | ±CI | AUC | ±CI | F1 Score | ±CI | Sensitivity | ±CI | Specificity | ±CI | PPV (precision) | ±CI | NPV | ±CI |
| MOSR | 0.664 | ±0.021 | 0.733 | ±0.026 | 0.720 | ±0.025 | 0.766 | ±0.050 | 0.679 | ±0.024 | 0.531 | ±0.036 | 0.766 | ±0.037 |
| Decision Tree Classifier | 0.614 | ±0.031 | 0.472 | ±0.028 | 0.648 | ±0.038 | 0.632 | ±0.060 | 0.668 | ±0.02 | 0.591 | ±0.035 | 0.632 | ±0.060 |
| LGBM Classifier | 0.680 | ±0.028 | 0.626 | ±0.033 | 0.727 | ±0.027 | 0.753 | ±0.051 | 0.705 | ±0.030 | 0.586 | ±0.067 | 0.753 | ±0.051 |
| Logistic Regression | 0.685 | ±0.022 | 0.667 | ±0.041 | 0.726 | ±0.023 | 0.741 | ±0.046 | 0.714 | ±0.022 | 0.612 | ±0.048 | 0.741 | ±0.046 |
| Random Forest Classifier | 0.701 | ±0.038 | 0.638 | ±0.043 | 0.750 | ±0.038 | 0.801 | ±0.076 | 0.710 | ±0.031 | 0.571 | ±0.073 | 0.801 | ±0.076 |
| XGB Classifier | 0.688 | ±0.042 | 0.621 | ±0.043 | 0.730 | ±0.040 | 0.749 | ±0.067 | 0.716 | ±0.039 | 0.610 | ±0.074 | 0.749 | ±0.067 |
| Clinical Scores | Accuracy | ±CI | AUC | ±CI | F1 Score | ±CI | Sensitivity | ±CI | Specificity | ±CI | PPV (precision) | ±CI | NPV | ±CI |
| ASA | 0.581 | | 0.630 | | 0.553 | | 0.459 | | 0.695 | | 0.739 | | 0.459 | |
| DASI | 0.563 | | 0.602 | | 0.720 | | 0.800 | | 0.563 | | 0.200 | | 0.950 | |
| CPET | 0.604 | | 0.751 | | 0.740 | | 0.970 | | 0.587 | | 0.093 | | 0.930 | |
| PPOSSUM | 0.563 | | 0.630 | | 0.720 | | 0.800 | | 0.563 | | 0.200 | | 0.950 | |
| Model – CRF-TS Dataset | Accuracy | ±CI | AUC | ±CI | F1 Score | ±CI | Sensitivity | ±CI | Specificity | ±CI | PPV (precision) | ±CI | NPV | ±CI |
| MOSR | 0.781 | ±0.056 | 0.850 | ±0.032 | 0.84 | ±0.037 | 0.824 | ±0.035 | 0.857 | ±0.038 | 0.782 | ±0.124 | 0.824 | ±0.054 |
| Decision Tree Classifier | 0.670 | ±0.041 | 0.661 | ±0.046 | 0.730 | ±0.044 | 0.684 | ±0.084 | 0.792 | ±0.042 | 0.640 | ±0.123 | 0.690 | ±0.084 |
| LGBM Classifier | 0.730 | ±0.053 | 0.748 | ±0.054 | 0.802 | ±0.041 | 0.832 | ±0.061 | 0.777 | ±0.043 | 0.530 | ±0.116 | 0.832 | ±0.061 |
| Logistic Regression | 0.690 | ±0.056 | 0.718 | ±0.075 | 0.767 | ±0.042 | 0.772 | ±0.057 | 0.766 | ±0.057 | 0.531 | ±0.162 | 0.772 | ±0.057 |
| Random Forest Classifier | 0.733 | ±0.048 | 0.764 | ±0.066 | 0.808 | ±0.033 | 0.851 | ±0.042 | 0.772 | ±0.046 | 0.503 | ±0.132 | 0.851 | ±0.042 |

*(Continued)*

**Table 2.** (Continued)

| Model – CRF and Clinical Dataset | Accuracy | ±CI | AUC | ±CI | F1 Score | ±CI | Sensitivity | ±CI | Specificity | ±CI | PPV (precision) | ±CI | NPV | ±CI |
|---|---|---|---|---|---|---|---|---|---|---|---|---|---|---|
| XGB Classifier | 0.709 | ±0.051 | 0.740 | ±0.066 | 0.788 | ±0.037 | 0.818 | ±0.055 | 0.762 | ±0.048 | 0.494 | ±0.138 | 0.818 | ±0.055 |
| **Model – CRF 585 subset** | **Accuracy** | **±CI** | **AUC** | **±CI** | **F1 Score** | **±CI** | **Sensitivity** | **±CI** | **Specificity** | **±CI** | **PPV (precision)** | **±CI** | **NPV** | **±CI** |
| MOSR | 0.695 | ±0.075 | 0.716 | ±0.086 | 0.783 | ±0.058 | 0.876 | ±0.067 | 0.709 | ±0.057 | 0.384 | ±0.145 | 0.876 | ±0.083 |
| Decision Tree Classifier | 0.565 | ±0.077 | 0.559 | ±0.059 | 0.656 | ±0.060 | 0.665 | ±0.082 | 0.650 | ±0.059 | 0.397 | ±0.136 | 0.665 | ±0.082 |
| LGBM Classifier | 0.621 | ±0.067 | 0.610 | ±0.069 | 0.710 | ±0.056 | 0.745 | ±0.082 | 0.682 | ±0.055 | 0.413 | ±0.137 | 0.745 | ±0.082 |
| Logistic Regression | 0.681 | ±0.072 | 0.701 | ±0.078 | 0.776 | ±0.050 | 0.884 | ±0.076 | 0.694 | ±0.053 | 0.342 | ±0.151 | 0.884 | ±0.076 |
| Random Forest Classifier | 0.620 | ±0.082 | 0.640 | ±0.090 | 0.718 | ±0.062 | 0.777 | ±0.089 | 0.672 | ±0.068 | 0.357 | ±0.178 | 0.777 | ±0.089 |
| XGB Classifier | 0.609 | ±0.077 | 0.608 | ±0.061 | 0.697 | ±0.072 | 0.728 | ±0.106 | 0.674 | ±0.059 | 0.409 | ±0.148 | 0.728 | ±0.106 |

Comparison of Multi-Objective Symbolic Regression (MOSR), versus Decision Tree Classifier, Light Gradient Bosting Machine (LGBM) Classifier, Logistic Regression, Random Forest Classifier and Extreme Gradient Boosting (XGB) Classifier. Values are expressed in mean and 95% CI from 10 execution on test set. CRF: Cardiorespiratory Fitness, CI: 95% Confidence interval, PPV: Positive Predictive Value, NPV: Negative Predictive Value, ASA: American Society of Anaesthesiologist score, DASI: Duke Activity Status Index, CPET: Cardiopulmonary Exercise Testing, PPOSSUM: Portsmouth Physiological and Operative Severity Score for the enumeration of Mortality and morbidity.

MOSR is an algorithm that distinguishes itself because of its flexibility, using Genetic Programming to create mathematical formulae for learning tasks. It stands out as it can utilize combinations of mathematical operations, covering both simple and complex functions, without adhering to a set model [19,20]. Another major advantage of MOSR is its capability to automatically identify the most crucial features during training. For this reason, we included all available variables in our experiments. For each experiment, we evolved a population of 300 individual models for 500 generations.

The ML library for MOSR is open source and available at: https://github.com/davideferrari92/multiobjective_symbolic_regression.

The Python package PyCaret was utilized to train Logistic Regression (LR), Decision Tree (DT), Random Forest (RF), Light Gradient Boosting (LGBM), and Extreme Gradient Boosting (XGB) for comparison against MOSR [25]. All models underwent training in three distinct experiments: the first focused exclusively on the clinical dataset, the second on the CRF dataset, and the final experiment utilized both databases. For hyperparameter optimization of the benchmark algorithms, we implemented grid search using PyCaret's default configuration. The hyperparameter search space was predefined, and the optimized parameters for each algorithm fell within this exploration range.

In accordance with TRIPOD guidelines, models underwent evaluation using established classification metrics, including accuracy, sensitivity, specificity, F1-Score, Positive Predictive Value (PPV – also defined precision), Negative Predictive Value (NPV), and Area Under the Curve (AUC) [26]. The results of the test set are reported in Table 2, the training test in S2 Table and the precision-recall curve for MORS in S3 Fig.

### Feature analysis

Shapley Additive exPlanations (SHAP) was applied to analyze contributions of the features included in the models. This method measures the impact of each individual feature enabling additional interpretation of relative contributions to final model performance [27,28].

### Clinical scores

ML model performance was benchmarked against existing clinical risk prediction scores, PPOSSUM, ASA and DASI. To predict binary outcomes of POMS 0-1 and POMS ≥2, thresholds were established for these scores. A threshold of

risk of 30% was set for PPOSSUM, while the ASA score threshold was set at a score of 2, and DASI at a score of 34 [29–32]. Logistic regression was performed using a CPET-derived ventilatory anaerobic threshold of 11 ml/$O_2$/kg/min ($VO_2$/Kg AT), a value commonly used to predict postoperative outcomes [33,34]. PPOSSUM was computed retrospectively for comparison. The Surgical Outcome Risk Tool (SORT) was omitted due to impractical retrospective physician assessment [35].

**Time-series (TS) features extraction and classification models**

The CRF-TS database, which includes data from 585 patients, was utilized to explore how CPET time-series data contribute to predicting patients with POMS 0-1 or POMS ≥2 on Postoperative Day 3. We experimentally rescaled the time-series to a lower dimension to reduce the complexity of the ML task without losing the informative shape of the curve; as shown in Fig 3A and 3B, the shape does not change when down sampled to 1000, 500 and 100. The MOSR model, as previously mentioned, was applied to this database, and its performance compared with that of LR, DT, RF, LGBM and XGB models. The robustness of these models was evaluated in a similar way to the previous experiments. Finally, we evaluated the models trained on the CFR-TS dataset against the same subset of patients using data from the CFR dataset (CRF 585 patient subset), employing the same algorithms for comparison (Fig 3C and 3D).

# Results

## Population description

1190 patients were included, with a median age of 71 years (61–79). Sixty-nine percent were male. The types of surgery reflect those routinely performed within our university hospital (Table 1). Intensive Care and Surgical High Dependency Units were the predominant initial postoperative destinations (64%). CPET data and key outcomes are shown in Table 1. Thirty-day mortality was 1.9% and one-year mortality 5.5%. A modest correlation existed between advancing age and an increased number of POMS-positive domains at day 3 (correlation coefficient: 0.27, $p < 0.05$), as indicated by the POMS score (Fig 1C). A similar, yet weaker, correlation existed for female sex (correlation coefficient: 0.091, $p < 0.05$). By contrast, $VO_2$/kg at ventilatory anaerobic threshold (AT) demonstrated a stronger inverse correlation with the number of POMS-positive domains on Postoperative Day 3 (correlation coefficient: 0.69, $p < 0.05$) (Fig 1D).

## MOSR models – Dataset comparison and SHAP analysis

The MOSR model incorporating only the clinical dataset, as presented in Fig 2A and Table 2, recorded the lowest metrics (AUC 0.733, F1-Score 0.720). Conversely, the MOSR model that exclusively used CRF data outperformed the others (AUC 0.946, F1-Score 0.890). The combined MOSR model, which included both CRF and clinical data, exhibited a performance slightly inferior to the model utilizing only CRF data (AUC 0.930, F1-Score 0.879). SHAP feature analysis of the MOSR model based on both CRF and clinical datasets highlights individual feature contributions to model performance and ranking, specifically lower $VO_2$peak and $VO_2$ at AT, elevated body mass index (BMI) and, finally, severity of the surgical procedure as being critical for model performance (Fig 2B).

## MOSR models – Comparison with clinical scores

Comparison of MOSR models against the clinical risk prediction scores (Fig 2C and Table 2) showed the MOSR CRF model outperformed all clinical scores (AUC 0.946, F1-score 0.890). Notably, the ASA (AUC 0.630, F1-score 0.553), DASI (AUC 0.602, F1-score 0.720) and PPOSSUM (AUC 0.630, F1-score 0.720) scores all had lower discriminative power compared to the MOSR models utilizing either CRF or clinical data. The model using an $VO_2$/Kg AT of 11 ml/kg/min as a discriminator achieved an AUC of 0.751 and F1-score of 0.740. Finally, the MOSR models outperformed other clinical scores in terms of PPV and NPV.

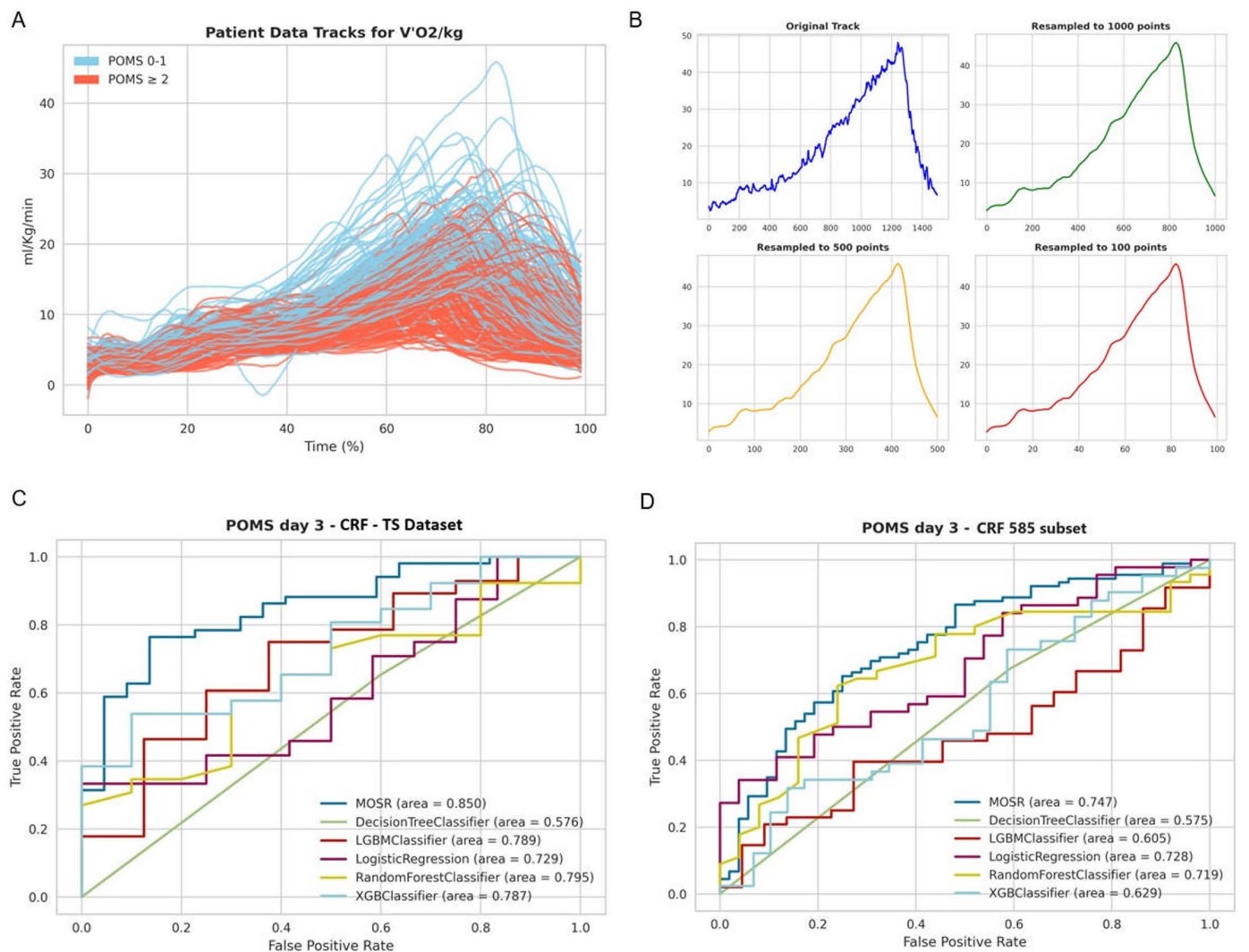

**Fig 3. A – Comparison of 585 time-series showing oxygen uptake per kilogram per minute (VO$_2$/Kg/min) in cardiopulmonary exercise testing (CPET).** The x-axis shows the moment of the exam, while the y-axis displays the VO$_2$/Kg/min values measured in ml/kg/min. Patients are categorized and color-coded based on their Postoperative Morbidity Survey (POMS) scores: those with POMS scores equal or less than 1 are represented in light blue, POMS scores exactly at or more than 2 in orange, and POMS scores greater than 3 in red. **B –** Illustration of the effect of resampling a data track on its resolution and detail. The top left plot shows the original track with high-resolution data points, depicted in blue. The other three plots demonstrate the track after resampling to different numbers of points, with each plot color-coded to represent a specific resampling: 1000 points in green (top right), 500 points in orange (bottom left), and 100 points in red (bottom right). **C -** Receiver Operating Characteristic (ROC) curves comparing the Multi-objective Symbolic Regression (MOSR) model against other machine learning classifiers: Decision Tree, LGBM Classifier, Logistic Regression, Random Forest Classifier, and XGB Classifier. Each curve represents the respective model's performance in predicting day 3 Postoperative Morbidity Survey (POMS) scores using Cardiorespiratory Fitness Time Series (CRF-TS) dataset. **D -** Receiver Operating Characteristic (ROC) curves comparing the Multi-objective Symbolic Regression (MOSR) model against other machine learning classifiers: Decision Tree, LGBM Classifier, Logistic Regression, Random Forest Classifier, and XGB Classifier. Each curve represents the respective model's performance in predicting day 3 Postoperative Morbidity Survey (POMS) scores using the subset of patient used in the Time-Series experiment from the Cardiorespiratory Fitness (CFR 585) dataset.

### MOSR model – Comparison with state-of-art ML models

Superior performance was exhibited by MOSR models compared to current state-of-the-art machine learning models, as indicated in Fig 2D–2F and Table 2. The CRF MOSR model outperformed the others (AUC 0.946, F1-score 0.890). The accuracy of the MOSR CRF model was 0.859 on the test set and 0.888 on the training set.

Consistent outcomes were observed when aggregating all CRF and clinical datasets, with the MOSR model surpassing others in performance (AUC 0.930, F1-score 0.879). The remaining models displayed marginally lower metrics. In the analysis using solely the clinical dataset, all models exhibited reduced performance relative to other tests. Overall, the MOSR models demonstrated higher NPV and PPV values compared to the other models.

The MOSR model showed a small difference in accuracy between the test (0.859) and training sets (0.886), whereas the gap was significantly larger for other models, The MOSR model led with the best performances (AUC 0.733, F1-score 0.720).

### Time-series MOSR model – Analysis and comparison

Using CRF from Time-Series (CRF-TS, Fig 3C), the MOSR model was the sole model to achieve performance metrics with an AUC of 0.85 and an F1-Score of 0.84. For the MOSR CRF-TS model, the accuracy was 0.781 on the test set and 0.783 on the training set, showing minimal difference. When the same subset of patients from the time-series were analyzed using the CRF dataset (CFR 585 subset), all models exhibited lower performances compared to the previously mentioned results (Fig 3D). The MOSR model achieved the highest performance (AUC 0.747, F1-Score 0.783). The main features utilized in the MOSR CRF-TS were $VO_2/HR$, VO2/Kg and $VCO_2/Kg$.

## Discussion

### Principal findings

We used a new machine learning method (called Multiobjective Symbolic Regression-MOSR), alongside other established Machine Learning models, to better predict which elderly patients might develop complications after a variety of major surgeries. Our approach builds on standard clinical information (like age, other illnesses, and lab results) by adding real data about a patient's fitness. We based our models on prospectively collected databases of comparable size to those used to develop the widely used Surgical Outcomes Risk Tool (SORT) [22,23]. Our findings highlight the critical role of CPET-derived cardiorespiratory fitness data in enhancing model performance over demographic and clinical features, underscoring the utility of formal evaluation of a patient's fitness prior to undergoing major surgery. We employed MOSR, an advanced and novel machine learning model, to develop predictive classifications that surpass the performance of state-of-the-art machine learning algorithms. Our models performed better than the clinical risk prediction scores PPOS-SUM, DASI and ASA, and could eventually be used to inform risk prediction for patients undergoing major surgery. We also described usage of SHAP analysis to explore the impact of individual features within ML models and to identify candidate mechanisms for future research. Furthermore, we detailed the innovative application of data obtained from a CPET time-series in training the MOSR; this surpasses the performance of manually extracted CPET data.

### The impact of physiology on predicting postoperative outcomes

Our analysis establishes a connection between $\dot{V}O_2$ peak, $\dot{V}O_2$ at AT, and $V_E/\dot{V}CO_2$ and postoperative outcomes, supporting findings from previous research [33,34] while also underscoring the significance of age in surgical outcomes. However, the link between Cardiorespiratory Fitness features ($\dot{V}O_2$ peak, AT and outcomes) was found to have greater influence than age on model performance. Linear regression alone proved insufficient for identifying patients at high risk for accumulating multiple POMS defined postoperative morbidities.

Our risk models, not specific to any type of surgery but considering operative severity, underscore that in an elderly population of patients with multiple comorbidities, the role of patient physiology and fitness rather than named comorbidity in

the development of perioperative complications. The best model, utilizing the MOSR algorithm and incorporating CRF data, demonstrated the best performance (Positive Predictive Value, or PPV increased by 49%). PPV tells us how many patients identified by models as "at risk" ended up having complications. For example, if using a hypothetical alternative method, if 100 patients were flagged as high risk, and only 20 truly experienced complications, a 49% improvement in PPV would mean that, with our new model, if you flagged 100 patients, roughly 30 of them (an extra 10 out of 100) would be at risk. In short, our model is much better at correctly identifying the patients who will experience problems, reducing the number of false positives. In addition to improvements in PPV, our model also showed a 20% improvement in NPV (negative predictive value), meaning it's better at identifying when a patient is not at risk (avoiding false negatives). Furthermore, overall accuracy measures (like AUC and F1-score) improved by 27% and 22%, respectively. These combined improvements help ensure that more patients are correctly classified, which could lead to better-tailored preoperative care through improved shared decision-making discussions and better risk data to support clinical decision making. In practical terms, this suggests that if incorporated into clinical risk prediction, the MOSR algorithm with cardiorespiratory fitness data could potentially improve diagnostic accuracy, leading to better identification of relevant cases and better-informed, personalized clinical decisions.

Cardiorespiratory fitness assessed through CPET is individual to the patient and likely reflects individual resilience. The results of this study indicate that the physiological response to acute stress (in this case exercise) is a better predictor of postoperative outcomes than traditional clinical diagnoses alone. This supports the incorporation of formal assessment of cardiorespiratory fitness in preoperative assessment when considering personalized risk assessment.

These findings are supported by the SHAP features analysis of the model utilizing the entire database, which reveals a significant impact of CPET parameters on the model's final performance. Notably, the most influential features were, in order, AT and $VO_2$ peak, ventilatory equivalents for $CO_2$ at AT and then body mass index, as described in the literature [36]. Although the severity of the operation was important, it played a lesser role. Therefore, the relationships and correlations between exercise response, patient body mass and composition, and the severity of surgery may lay the groundwork for future research.

MOSR and other machine learning models surpassed clinical risk prediction scores PPOSSUM, ASA and DASI scores by roughly 30%, and the AT-based regression model by approximately 20%. The modest effectiveness of PPOSSUM and DASI observed in this study aligns with findings from other recent studies [37–39]. Additionally, the ultimate MOSR fitness model showed promising results when compared to the Surgical Outcome Risk Tool (SORT) for morbidity (AUROC 0.72, 95% CI: 0.67-0.77), which was developed using a population that overlaps with the UCLH CPET database [23].

### CPET time-series data for CRF fitness

Most CPET studies have traditionally focused on a narrow set of individual non-dynamic measures, including $VO_2$ peak, AT, and VE:$VCO_2$ at AT, due to their predictive power and physiological relevance. However, this approach overlooks most data obtainable from CPET. Our time-series based models emphasized the significance of dynamic exercise responses. Training the models on to the same subset of patients, the model using time-series data achieved, on average, a 12% increase in AUC and a 7% improvement in F1-score compared to models developed with hand-extracted CPET data on the population. These data could be used to enhance our comprehension of how various elements of the physiological response to exercise contribute to resilience against surgical trauma. Interestingly, the primary features utilized by the MOSR model are derived from the $VO_2$/HR, VO2/Kg, and $VCO_2$/Kg time-series. This highlights the significance of cardiac output and heart rate adaptation, lending physiological credibility to the model's responses and supporting the concept of cross-stressor adaptation [15,40,41].

### MOSR performances and model interpretation

MOSR consistently outperformed all other algorithms and, analyzing the features included in the final models, we can value its intrinsic feature selection behavior. Moreover, in contrast to all other ML approaches that rely on Binary Cross

entropy for classification training, MOSR was trained by also optimizing for the Partial Akaike Information Criterium [42] that penalizes high complexity models, effectively maximizing the ratio between performance and model complexity. In our experiments, MOSR demonstrated excellent predictive performance and generalization capability; most importantly, it did not overfit the training data like other algorithms, making it the most favorable choice. Particularly in models with CRF derived from time-series data, MOSR produced the best AUC and F1-scores on the test set, proving to be the most reliable algorithm out of our selection. In our study the potential of MOSR for correlating patient clinical and physiological features with perioperative outcomes is highlighted, showcasing its capability to autonomously select important features and create models that are both simple and accurate, demonstrating strong generalizability.

## Limitations and strengths

This study represents the first investigation using a perioperative CRF fitness dataset with ML to predict postoperative outcomes in an elderly population. The study's primary limitation stems from being conducted at a single center and its retrospective nature. However, the 28-day mortality after elective non-cardiac major surgery in our institution (2%) is comparable with the wider literature (1.1 to 3.6%), implying that our population is not significantly different from that in other comparable centers [5,43]. Given encouraging initial results, there is an intention to carry out prospective external validation. The strength of our work lies in usage of POMS-defined morbidity and the usage of MOSR to moderate traditional limitations of ML techniques such as overfitting to single datasets and its incumbent capacity for easy external translation to external datasets.

## Conclusion

We have demonstrated that incorporating CPET-derived CRF fitness data into a machine learning model, specifically MOSR, can significantly improve preoperative risk prediction in elderly surgical patients compared to traditional clinical risk scores. These data offer deeper insights compared to the clinical data conventionally used for risk prediction. This approach could enable clinicians to better identify patients at high risk of postoperative complications, allowing for targeted interventions and potentially reducing morbidity. Additionally, leveraging data directly from the CPET time-series appears to enhance model performance, underscoring the value of examining elder patient physiology and adaptability to acute stress. We also introduced a novel ML model, MOSR, to the clinical field, which outperformed existing state-of-the-art ML models. These findings lay the groundwork for developing clinical decision support tools that could optimize postoperative care pathways for high-risk elderly surgical patients and advance precision medicine, potentially improving patient care and reducing hospital budgets [44–46].

## Supporting information

**S1 Fig. CONSORT flow diagram illustrating the selection process of patients for the study, demonstrating final sample sizes for tabular and time-series analysis.**
(TIF)

**S2 Fig. Linear regression between number of POMS positivity at Day 3, Day 5, and Day 7.**
(TIF)

**S3 Fig. PR curves for the Multi-objective Symbolic Regression (MOSR) models: A - Clinical Dataset, B - CFR Dataset, C - Combined CFR and Clinical Dataset.**
(TIF)

**S1 Table. Postoperative Morbidity Score (POMS), adapted from [22].**
(DOCX)

**S2 Table. Machine learning model performances TRAINING set.** Comparison of Multi-Objective Symbolic Regression (MOSR), versus Decision Tree Classifier, Light Gradient Bosting Machine (LGBM) Classifier, Logistic Regression, Random Forest Classifier and Extreme Gradient Boosting (XGB) Classifier. Values are expressed in mean and 95% CI from 10 execution on training set. CRF: Cardiorespiratory Fitness, CI: 95% Confidence interval, PPV: Positive Predictive Value, NPV: Negative Predictive Value, ASA: American Society of Anaesthesiologist score, DASI: Duke Activity Status Index, CPET: Cardiopulmonary Exercise Testing, PPOSSUM: Portsmouth Physiological and Operative Severity Score for the enumeration of Mortality and morbidity.
(DOCX)

## Acknowledgments

Pietro Arina is supported by funds from the Cleveland Clinic London Hospital, London, United Kingdom and by the Mittal Fund at Cleveland Clinic Philanthropy (UK). Evangelos B. Mazomenos is supported by the Wellcome Trust and Engineering and Physical Sciences Research Council under grant Nos. EP/Z534754/1, 203145Z/16/Z and NS/A000050/1. Davide Ferrari is funded and supported by King's College London and DRIVE-Health, KCL funded Centre for Doctoral Training (CDT) in Data-Driven Health. John Whittle is supported by funds from the University College London Hospitals National Institute of Health Research Biomedical Research Centre Critical and Perioperative Care theme and in part by an International Anesthesia Research Society Mentored Research Grant.

## Author contributions

**Conceptualization:** Pietro Arina, Ramani Moonesinghe, Mervyn Singer, John Whittle, Evangelos B. Mazomenos.

**Data curation:** Pietro Arina, Davide Ferrari, Nicholas Tetlow, Amy Dewar, Robert Stephens, Daniel Martin, Ramani Moonesinghe, Mervyn Singer, John Whittle, Evangelos B. Mazomenos.

**Formal analysis:** Pietro Arina, Davide Ferrari, Maciej R. Kaczorek, Nicholas Tetlow, Robert Stephens, Ramani Moonesinghe, Mervyn Singer, John Whittle, Evangelos B. Mazomenos.

**Funding acquisition:** Pietro Arina, Ramani Moonesinghe, Mervyn Singer, John Whittle, Evangelos B. Mazomenos.

**Investigation:** Pietro Arina, Davide Ferrari, Maciej R. Kaczorek, Nicholas Tetlow, Amy Dewar, Daniel Martin, Ramani Moonesinghe, Mervyn Singer, John Whittle, Evangelos B. Mazomenos.

**Methodology:** Pietro Arina, Davide Ferrari, Maciej R. Kaczorek, Nicholas Tetlow, Daniel Martin, Ramani Moonesinghe, Mervyn Singer, John Whittle, Evangelos B. Mazomenos.

**Project administration:** Pietro Arina, Mervyn Singer, John Whittle, Evangelos B. Mazomenos.

**Resources:** Pietro Arina, Amy Dewar, Robert Stephens, Mervyn Singer, John Whittle, Evangelos B. Mazomenos.

**Software:** Pietro Arina, Davide Ferrari, Maciej R. Kaczorek, Mervyn Singer, John Whittle, Evangelos B. Mazomenos.

**Supervision:** Pietro Arina, Davide Ferrari, Ramani Moonesinghe, Mervyn Singer, John Whittle, Evangelos B. Mazomenos.

**Validation:** Pietro Arina, Davide Ferrari, Nicholas Tetlow, Ramani Moonesinghe, Mervyn Singer, John Whittle, Evangelos B. Mazomenos.

**Visualization:** Pietro Arina, Davide Ferrari, Maciej R. Kaczorek, Ramani Moonesinghe.

**Writing – original draft:** Pietro Arina, Davide Ferrari, Nicholas Tetlow, Robert Stephens, Daniel Martin, Ramani Moonesinghe, Mervyn Singer, John Whittle, Evangelos B. Mazomenos.

**Writing – review & editing:** Pietro Arina, Nicholas Tetlow, Amy Dewar, Ramani Moonesinghe, Mervyn Singer, John Whittle, Evangelos B. Mazomenos.

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
