## [Decision Letter · Decision Letter 0]

13 Jan 2025

PDIG-D-24-00447Assessing perioperative risks in a mixed elderly surgical population: a multi-objective symbolic regression approach to cardiorespiratory fitness derived from cardiopulmonary exercise testingPLOS Digital Health Dear Dr. Arina, Thank you for submitting your manuscript to PLOS Digital Health. After careful consideration, we feel that it has merit but does not fully meet PLOS Digital Health's publication criteria as it currently stands. Therefore, we invite you to submit a revised version of the manuscript that addresses the points raised during the review process. Please submit your revised manuscript within 60 days Mar 14 2025 11:59PM. If you will need more time than this to complete your revisions, please reply to this message or contact the journal office at digitalhealth@plos.org.  Please include the following items when submitting your revised manuscript:* A rebuttal letter that responds to each point raised by the editor and reviewer(s). You should upload this letter as a separate file labeled 'Response to Reviewers '. This file does not need to include responses to any formatting updates and technical items listed in the 'Journal Requirements' section below.* A marked-up copy of your manuscript that highlights changes made to the original version. You should upload this as a separate file labeled 'Revised Manuscript with Track Changes '.* An unmarked version of your revised paper without tracked changes. You should upload this as a separate file labeled 'Manuscript '. If you would like to make changes to your financial disclosure, competing interests statement, or data availability statement, please make these updates within the submission form at the time of resubmission. Guidelines for resubmitting your figure files are available below the reviewer comments at the end of this letter. We look forward to receiving your revised manuscript. Kind regards, Dhiya Al-Jumeily OBE, PhDSection EditorPLOS Digital Health  Leo Anthony CeliEditor-in-ChiefPLOS Digital Healthorcid.org/0000-0001-6712-6626 **Journal Requirements:** 

2. Please provide an Author Summary. This should appear in your manuscript between the Abstract (if applicable) and the Introduction, and should be 150–200 words long. The aim should be to make your findings accessible to a wide audience that includes both scientists and non-scientists. Sample summaries can be found on our website under Submission Guidelines:

https://journals.plos.org/globalpublichealth/s/submission-guidelines#loc-parts-of-a-submission.

**Additional Editor Comments (if provided):**  Revise please considering the reviewers comments. Specific attention should be paid to the writing of the paper and the citations. Please cite statements where relevant and revise citations and references.**Reviewers' Comments:** Reviewer #1: Manuscript #: PDIG-D-24-00447

Title: Assessing perioperative risks in a mixed elderly surgical population: a multi-objective symbolic regression approach to cardiorespiratory fitness derived from cardiopulmonary exercise testing

By Pietro Arina et al.

The authors explored the potential of integrating cardiorespiratory fitness (CRF) data from cardiopulmonary exercise testing (CPET) into machine learning (ML) models to improve the accuracy of preoperative risk assessment for predicting postoperative morbidity in elderly patients undergoing major elective surgery. The study found that incorporating CRF data significantly enhanced the performance of all models, with the Multi-Objective Symbolic Regression (MOSR) model, a novel algorithm based on genetic programming, consistently outperforming traditional risk scores and other ML approaches. The authors also examined the impact of using CPET time-series data directly, demonstrating a further improvement in model performance over models based on manually extracted CPET features. The findings suggest that incorporating CPET-derived CRF data into ML models has the potential to revolutionize perioperative risk prediction in elderly surgical patients, potentially leading to more targeted interventions and improved patient outcomes.

The use symbolic regression approach to model the CRF data is a major strength. The manuscript, however, is written in a rush, with many glaring errors, which renders the paper quality extremely poor. The overall findings are interesting but can be further improved. The model discovered by symbolic regression needs some elaborations/explanations for the variables involved, perhaps some links to the SHAP features. This would be more interesting than simple prediction. Overall, figure legends are way too small to read. Figure quality and some explanations need improvement. Other specific comments are as follows:

1. Of 2,145 patients assessed, 1,190 were included in the tabular database, and 585 with full CPET recordings for time-series analysis, but in Supplementary Figure 1, by Excluded n=1005, there are only 1140, not 1190 patients. The patient number does not match.

2. What are the purposes to have Major and Minor POMS morbidity classification?

3. Which software is used for MOSR analysis? It would be good to make the code public available, and even better if you could make the data available.

4. In the Abstract, it says “A model was also developed for the same task using data directly extracted from the CPET time-series.”. Which model do you refer to? And what are the conclusions of this model?

5. The reference in the paper are terribly messed up, some places using authors, year e.g. the Lee et al., 2022; J. Rose et al., 2015; Lee et al., 2022)) , but other place using number, fpr example, [25]. There are many problems: 1. Repeated reference: “Lee et al., 2022; J. Rose et al., 2015; Lee et al., 2022))”; 2. “DenckerTjeertes et al., 2021)2016)” unsure what it is; 3. Ferrari, Guidetti, Wang, et al., 2022b2022), et c etc …You really need careful read and check before submission. It’s a good material, that’s why I do not reject the paper.

6. Not complete sentence, “dherence to Caldicott principles ensured data confidentiality and proper collection methods”, no period at the end of sentence.

7. Instead of simply listing the statistics used, “statistical tests such as Student’s t-test, Mann-Whitney U test, Chi-Squared test, and Fisher’s Exact Test were utilized.”, I suggest to explicitly indicate each test when used.

8. Why testing runs on a random 90% sample (214) of the testing set, but not the whole testing data?

9. “e evolved a population of 300 individual models for 500 generations.”, why 300 and 500 used?

10. Why the model performance with more information by using both CRF and clinical data gets worse than using either data alone?

11. Fig.2c, “Cardiopulmonary Exercise Testing (CPET) (red),”, there is no red curve.

Reviewer #2: Assessing perioperative risks in a mixed elderly surgical population

Introduction

The first sentence is a little awkward, suggest revising.

“Accurate preoperative risk prediction supports patients in making informed decisions and guides clinical decision-making.” Is risk prediction for the patient or clinican?

Try to limit abbreviations; a number of times I had to go back to find what some meant

Methods

There is no mention of inclusion/exclusion criteria in text. How were the patients selected? Particularly those that were included.

There does not appear to be reference to a clinical trial registration. Please include.

How were ventilatory equivalents reported? Values at the ventilatory threshold, at max?

How was the POMS administered? In hospital/home? Free of people/distractions or was it supervised by a researcher?

Table 1 – “CPET values” – these just appear to be resting physiological variables independent of CPET.

What is the MET value here? You have not described how it is determined in methods. It does not appear to equate to the peak VO2?

“VE/V̇CO2 (ml/min)” – VE/VCO2 doesn’t have a unit as far as I am aware as it is a ratio.

Results

“A modest correlation existed between advancing age and an increased

number of POMS-positive domains at day 3, as indicated by the POMS score (Figure-1C)” It would be helpful here and else where to report the correlation (with 95% CI).

Discussion

It would be helpful to explain some of your complex statistical results, in a more lay format. For example, what does an improvement of 49% for PPV mean? How many more patients are accurately captured or how many are still missed?

“The best model, utilizing the MOSR algorithm and incorporating CRF data, demonstrated an average increase of 27% in AUC and a 22% improvement in F1-score” What does this mean for a clinician, in practical terms?

The term “fitness response” does not make sense…do you mean CRF or exercise response

Reviewer #3: I am concerned about the accuracy of the model. The AUC values are very high which makes me questions the inputs. Are there too many overlapping inputs? Consider splitting some of the dataset in the inputs to see which one is degrading the outcomes. The SHAP plots and some of the figures need revision.

---

## [Editor Report · Decision Letter 1]

5 Apr 2025

Assessing perioperative risks in a mixed elderly surgical population: a multi-objective symbolic regression approach to cardiorespiratory fitness derived from cardiopulmonary exercise testing

PDIG-D-24-00447R1

Dear Dr Arina,

We are pleased to inform you that your manuscript 'Assessing perioperative risks in a mixed elderly surgical population: a multi-objective symbolic regression approach to cardiorespiratory fitness derived from cardiopulmonary exercise testing' has been provisionally accepted for publication in PLOS Digital Health.

Best regards,

Dhiya Al-Jumeily OBE, PhD

Section Editor

PLOS Digital Health